# LKB1 Differently Regulates Adipogenesis in Intramuscular and Subcutaneous Adipocytes through Metabolic and Cytokine-Related Signaling Pathways

**DOI:** 10.3390/cells9122599

**Published:** 2020-12-04

**Authors:** Ziye Xu, Yanbing Zhou, Qiuyun Nong, Wenjing You, Liyi Wang, Yizhen Wang, Tizhong Shan

**Affiliations:** 1College of Animal Sciences, Zhejiang University, Hangzhou 310058, China; ziyexu@zju.edu.cn (Z.X.); 11817022@zju.edu.cn (Y.Z.); qynong@zju.edu.cn (Q.N.); youwj@zju.edu.cn (W.Y.); 21917014@zju.edu.cn (L.W.); chuxn@zju.edu.cn (Y.W.); 2Key Laboratory of Molecular Animal Nutrition (Zhejiang University), Ministry of Education, Hangzhou 310058, China; 3Key Laboratory of Animal Feed and Nutrition of Zhejiang Province, Hangzhou 310058, China

**Keywords:** LKB1, adipogenesis, subcutaneous fat, intramuscular fat, adipocyte, transcriptomics

## Abstract

Liver kinase B1 (LKB1) plays important and various roles in the differentiation and lipid metabolism of adipocytes. However, the current knowledge of the respective roles of LKB1 in subcutaneous fat (SCF) and intramuscular fat (IMF) adipocytes remains unclear. This study aimed to discover the different regulatory mechanisms of LKB1 in SCF and IMF adipocytes. We found that LKB1 overexpression inhibited adipogenesis in both SCF and IMF adipocytes, and SCF adipocytes were more sensitive to regulation by LKB1. Transcriptomics results showed that IMF adipocytes had many more differentially expressed genes (DEGs) than SCF adipocytes. Pathway analysis of the shared and distinct DEGs revealed that the main adipogenesis mechanism was similar between SCF and IMF adipocytes upon LKB1 overexpression, while regulatory and metabolic signaling pathways, such as MAPK, PPAR signaling pathways, were differently regulated by LKB1. Several cytokine-related pathways were only enriched in LKB1-overexpressing IMF adipocytes. Our study reveals different regulators and signaling pathways between SCF and IMF adipocytes under LKB1 overexpression, which may be potential targets to differentially control SCF and IMF deposition and improve our understanding of the regulatory mechanisms of IMF deposition.

## 1. Introduction

In humans, both subcutaneous fat (SCF) and intramuscular fat (IMF) are reported to be associated with several metabolic dysfunctions, including insulin resistance, obesity, and diabetes [1,2]. In farm animals, however, the IMF content is a crucial factor that positively related to the tenderness, juiciness and flavor of meat and is considered to be of great value in improving meat quality, while SCF affects the lean meat ratio of the carcass and reduces its economic value [3,4,5]. Thus, uncovering the metabolic features and regulatory mechanism of IMF development is of both great medical value and economic value. Although both IMF and SCF are mainly composed of white adipocytes, evidence has now accumulated to indicate that their properties differ in many aspects, such as proliferation, differentiation, lipogenic capacity, as well as lipolytic activity, fatty-acid oxidation and basal-energy metabolism [6,7,8]. These differences may be caused by the different manners of glucose utilization, lipid metabolism and regulatory signaling pathways in IMF and SCF [9,10].

Liver kinase B1 (LKB1), also called serine/threonine-protein kinase 11 (STK11), is a master kinase that was first described as a tumor suppressor gene that plays critical roles in various cellular processes by phosphorylating and activating kinases of the AMP-activated protein kinase (AMPK) subfamily [11]. Previous studies demonstrated that LKB1 is required for adipose tissue development and thermogenesis [12,13]. In preadipocytes, LKB1 influences adipogenesis and lipogenesis by regulating the expression of the master adipogenesis regulator CCAAT/enhancer-binding protein alpha (C/EBPα) and peroxisome proliferator activated receptor gamma (PPARγ) in 3T3-L1 preadipocytes or mouse embryonic fibroblasts (MEFs) [12,14]. However, the exact effect of silencing or deleting LKB1 in preadipocytes is still controversial [12,14]. Our previous study demonstrated that adipocyte-specific deletion of LKB1 marginally promoted brown adipose tissue (BAT) and white adipose tissue (WAT) stromal vascular fraction (SVF) cell differentiation, lipid accumulation, and thermogenesis [13]. Conversely, overexpression of LKB1 inhibited BAT and WAT SVF cell differentiation and lipid accumulation [13]. LKB1 overexpression in 3T3-L1 adipocytes also inhibited adipogenesis and induced lipid catabolism [15]. Overall, LKB1 signaling plays important and various roles in WAT and BAT. However, whether LKB1 can differentially regulate adipogenesis and lipid metabolism of SCF and IMF adipocytes is still unclear

We hypothesized that LKB1 could differentially regulate adipogenesis and lipid metabolism in SCF adipocytes and IMF adipocytes. In the present study, SCF and IMF preadipocytes obtained from mice or pigs were induced to differentiate into mature adipocytes and followed by LKB1 overexpression. The mRNA and protein expression levels of genes related to adipogenesis, lipogenesis and lipid metabolism were detected by quantitative real-time PCR, Western blot and transcriptomics, respectively. Our study helps to elucidate the difference between SCF and IMF preadipocytes during differentiation and upon LKB1 overexpression, revealing various different regulatory factors and pathways between SCF and IMF preadipocytes, which may be potential targets to differentially control SCF and IMF deposition.

## 2. Materials and Methods

### 2.1. Cell Isolation and Culture

All procedures involving mice and pigs were approved by the Zhejiang University Animal Care and Use Committee. The ethical committee number for the study is ZJU20160346. The pig and mouse IMF and SCF preadipocytes were isolated and cultured as described previously [16,17,18]. Briefly, subcutaneous white fat and longissimus dorsi (LD) muscles were harvested from three 3-day-old male Duroc × Landrace × Yorkshire (DLY) pigs. Inguinal white adipose tissue (iWAT) and tibial anterior (TA) muscle were removed from three 2-week-old male C57BL/6 mice for each batch. To avoid the individual difference and batch effect, the mouse and pig IMF and SCF preadipocytes for parallel groups in each experiment are isolated from the same individuals at the same time. Harvested fat and muscle were minced into small pieces and digested with 0.15% collagenase type I (Life Technologies Corporation, Grand Island, USA) in DMEM/F12 (TBD science, Tianjin, China) solution at 37 °C for 1.5 h and 0.5 h, respectively. The digested samples were filtered with 70 μm and 100 μm cell strainer, and cells were washed twice with phosphate-buffered saline (PBS) by centrifugation at 200× *g* for 10 min. Finally, cells were transferred to plates and grown in growth medium (GM) (DMEM/F12 supplemented with 15% fetal bovine serum (FBS) (Life Technologies Corporation) and 1% penicillin–streptomycin) at 37 °C with 5% CO_2_. After 2 h, the nonadherent cells were washed, and the medium was replaced with fresh medium to continue culturing.

After reaching 90% confluence (about 48 h), IMF and SCF preadipocytes were induced to differentiate with the adipogenic differentiation agent, MDI (0.5 mM 3-isobutyl-1-methylxanthine, 1 μM dexamethasone, and 10 mg/L insulin), for 3 days. The medium was replaced with maintenance medium containing 1 mg/L insulin and 10% FBS–DMEM until day 5, and fresh medium was added every 2 days.

### 2.2. Establishment of LKB1-Overexpressing Cells

The LKB1-overexpressing adenovirus and control GFP-adenovirus were gifts from Jiangang Gao (Institute of Developmental Biology, School of Life Science, Shandong University, Jinan, Shandong, China). For the adipogenesis related experiments (4~6 replicates per treatment), IMF and SCF preadipocytes were infected with control GFP-adenovirus or LKB1-overexpression adenovirus at the same time as MDI medium application. For RNA-seq analysis, two days before harvesting, IMF and SCF preadipocytes (3 replicates per treatment) were infected by control GFP-adenovirus or LKB1-overexpressing adenovirus according to the manufacturer’s protocol.

### 2.3. Flow Cytometric Analysis

A flow cytometry assay was used to characterize intramuscular and subcutaneous preadipocytes for the expression of preadipocyte-specific surface marker Sca1 [19]. Trypsin was used to remove the cells from the culture flasks. The cells were then pelleted by centrifugation. Flow cytometry buffer (FCB; 1× DPBS, FBS 2%) was used to wash the cells. The cell suspensions were then aliquoted and incubated with anti-Sca1 antibody (BD Pharmingen™, Becton Dickinson and Company, USA) for 30 min. Flow cytometry analysis was conducted on a BD FACSVerse™ flow cytometer (Becton Dickinson and Company, Franklin Lakes, NJ, USA).

### 2.4. Oil-Red-O Staining

To quantify lipid accumulation in IMF and SCF preadipocytes, Oil Red O staining was performed on day 5 of adipogenic differentiation [20]. Briefly, the adipocytes were washed twice with PBS, fixed with 10% phosphate-buffered formalin for 15 min. After fixation, the cells were washed three times with PBS and stained with Oil Red O working solution for 30 min at room temperature, and then washed for 20 s with 60% isopropanol to remove the excess stain. The stained lipid droplets within adipocytes were visualized by Leica Digital Firewire Color Cameras (DFC420, Leica Microsystems Ltd., Wetzlar, Germany).

### 2.5. Protein Isolation and Western Blotting

Total protein extracts were isolated from harvested cells using RIPA buffer supplemented with protease inhibitor cocktail (Thermo Fisher Scientific, Waltham, MA, USA). The protein concentration was determined with the BCA protein assay (Thermo Fisher Scientific, USA). A total of 20 μg protein from each sample was separated by SDS–PAGE and transferred to polyvinylidene fluoride (PVDF) membranes (GVS North America, Sanford, FL, USA). The primary antibodies used were against FABP4 (HuaAn Biotechnology Co., Ltd., Hangzhou, China), LKB1 (sc-32245 Santa Cruz Biotechnology Inc., Dallas, TX, USA), and GAPDH (HuaAn Biotechnology Co., Ltd., Hangzhou, China). Primary antibodies were detected using anti-rabbit or anti-mouse horseradish peroxidase (HRP)-conjugated IgG (Thermo Fisher Scientific, USA). Signals were visualized using enhanced chemiluminescence (Beijing Labgic Technology Co., Ltd., Beijing China).

### 2.6. Quantitative Real-Time PCR (qPCR)

Total RNA was lysed from harvested cells using TRIzol reagent (Yeasen Biotech Co., Ltd., Shanghai, China). We used a Hifair^®Ⅱ^1st Strand cDNA Synthesis SuperMix (Yeasen Biotech Co., Ltd., Shanghai, China) to reverse-transcribe 1.0 μg of each sample to the first-strand cDNA. The mRNA expression levels of related genes were detected by Synergy brands (SYBR) green dye qPCR (Yeasen Biotech Co., Ltd., Shanghai, China) with the CFX96 Touch Real-Time PCR detection system (Bio-Rad, Berkeley, CA, USA) according to the manufacturer’s instructions. 18S rRNA was used as an internal control. The primer sequences used are provided in Appendix A [21].

### 2.7. RNA Isolation, Library Construction and RNA-Seq Analysis

RNA extraction, library construction and RNA-seq analysis of harvested cells were performed by Sangon Biotech using previously published methods [22]. Total RNA was extracted using the Total RNA Extractor (TRIzol) Kit (B511311, Sangon, China) according to the manufacturer’s protocol and treated with RNase-free DNase I to remove genomic DNA contamination. A total amount of 2 μg RNA per sample was used as input material for subsequent RNA sample preparations. Sequencing libraries were generated using the VAHTSTM mRNA-seq V2 library prep kit for Illumina^®^, following the manufacturer’s recommendations, and index codes were added to attribute sequences to each sample. The libraries were then quantified and pooled. Paired-end sequencing of the library was performed on HiSeq XTen sequencers (Illumina, San Diego, CA, USA). The quality of the sequenced data was evaluated by FastQC (version 0.11.2). Trimmomatic (version 0.36) was applied to filter raw reads, and clean reads were mapped to the reference genome by HISAT2 (version 2.0) with the default parameters. RSeQC (version 2.6.1) was used to run statistics on the alignment results. The homogeneity distribution and the genomic structure were calculated by Qualimap (version 2.2.1). BEDTools (version 2.26.0) was applied for statistical analysis of the gene coverage ratio. StringTie (version 1.3.3b) was used to compute the gene expression values of the transcripts. Transcripts per kilobase million (TPM), which eliminated the influence of gene lengths and sequencing discrepancies, was applied for direct comparison of gene expression between samples. The package DESeq2 (version 1.12.4) in R software (version 3.5.1) was used for the gene count data to determine differentially expressed genes (DEGs) between the two groups. Genes with values <0.05 and |log2 (fold change) | >0.5 were considered significant DEGs. The functions ggplot2(), pheatmap() and cor() in R software were applied to generate scatter plot graphs, heatmaps and correlation graphs, respectively.

### 2.8. Pathway Enrichment Assay

DEGs were subjected to gene ontology (GO) functional analysis and Kyoto Encyclopedia of Genes and Genomes (KEGG) pathway analysis using the packages clusterProfiler and org.Ss.eg.db in R software (version 3.5.1) [23]. DEGs were mapped to the GO terms (biological functions) in the database. The number of genes in every term was calculated, and a hypergeometric test was performed to identify significantly enriched GO terms in the gene list out of the background of the reference-gene list. Enriched terms were visualized by a barplot and cnetplot. The network of most enriched terms was visualized by cnetplot. KEGG pathway analysis identified significantly enriched metabolic pathways or signal transduction pathways enriched in DEGs compared to those of a reference gene background using the hypergeometric test. Enriched pathways were visualized by dotplot.

### 2.9. Statistical Analysis

The data of carcass and meat characteristics, enzyme activities and fatty acid components are presented as the mean ± SEM. Comparisons were made by unpaired two-tailed Student’s *t*-tests. Differences among groups were considered statistically significant at *p* < 0.05.

## 3. Results

### 3.1. Overexpression of LKB1 Inhibits Pig IMF and SCF Preadipocytes Differentiation

IMF and SCF were isolated from the LD muscle and the SAT of 3-day-old pigs and digested with collagenase Ⅰ to separate pig IMF and SCF preadipocytes (Figure 1A). qPCR results showed dramatic increases in the mRNA of adipogenic master regulators (*Pparg*, *Cebpa*) and mature adipocyte markers (*Adipoq*, *Fabp4*) after adipogenic differentiation in both SCF (1041.9%, 683.7%, 22,181.2%, 16,613.0%, respectively) and IMF (3801.3%, 618.3%, 142,345.9%, 328,148.7%, respectively) adipocytes (Figure 1B,C). IMF preadipocytes expressed lower levels of *Pparg*, *Adipoq* and *Fabp4* than SCF preadipocytes, which were reversed in mature adipocytes after adipogenic differentiation (Figure 1B,C). The mRNA of *Leptin* was decreased after adipogenic differentiation in both SCF (−54.6%) and IMF (−55.9%) adipocytes, and IMF adipocytes expressed lower levels of *Leptin* than SCF adipocytes both before and after adipogenic differentiation (Figure 1D). In addition, the mRNA of *Lkb1* was also decreased after adipogenic differentiation in both SCF and IMF adipocytes, and the *Lkb1* expression level showed an even steeper decrease in SCF adipocytes (68.2%) than IMF adipocytes (19.7%) (Figure 1E). Thus, we concluded that LKB1 might play a role in differently regulating intramuscular fat and subcutaneous fat content in pigs.

Next, we generated LKB1-overexpressing IMF and SCF adipocytes and examined the mRNA levels of adipogenic genes. The mRNA level of *Lkb1* dramatically increased both in LKB1-overexpressing SCF and IMF adipocytes (27,387%, 473,255%, respectively) and elevated even more in IMF adipocytes (Figure 1F). The mRNA levels of *Pparg* and *Cebpa* were remarkably decreased both in LKB1-overexpressing IMF and SCF adipocytes and reduced even more in SCF adipocytes (−71.7%, −69.1%, respectively) than in IMF adipocytes (−46.7%, −20.1%, respectively) (Figure 1G). The mRNA levels of lipolysis related genes (*Hsl*, *Atgl*) were notably decreased to a similar degree in LKB1-overexpressing IMF (−79.2%, −76.0%, respectively) and SCF (−78.1%, −67.1%, respectively) adipocytes (Figure 1H). There was an increasing trend in the mRNA of *Leptin* in both LKB1-overexpressing IMF (175%) and SCF (76.3%) adipocytes (Figure 1I). The mRNA level of glucose metabolism regulator, *Glut4*, was significantly reduced by LKB1 overexpression in both IMF (−86.7%) and SCF adipocytes (−59.7%) (Figure 1J). These results suggested that LKB1 may play a role in differentially regulating adipogenesis and glucose metabolism in IMF and SCF adipocytes from pigs.

### 3.2. LKB1 Overexpression Alters the Transcriptional Profile of Pig SCF Adipocytes

To explore how the transcriptional profile changed in SCF adipocytes upon LKB1 overexpression, we performed RNA-seq to map the transcriptional changes and adipogenic regulatory pathways in LKB1-overexpressing SCF adipocytes and controls. We identified a total of 1247 DEGs, of which 655 were upregulated, and 592 were downregulated by LKB1 overexpression (Figure 2A). The mRNA level of LKB1 was dramatically elevated 8.4 times in LKB1-overexpressing SCF adipocytes compared to that in controls (Figure 2A). GO enrichment analysis of DEGs induced by LKB1 overexpression revealed pronounced changes in biological processes (mitotic cell cycle process, nuclear division), molecular functions (microtubule motor activity, DNA-binding transcription factor activity) and cell components (condensed chromosome, spindle) (Figure 2B and Appendix A). The cnetplot depicted the linkages of three significant GO terms in the biological process category (mitotic cell cycle process, mitotic cell cycle, nuclear division) and genes involved in these terms to generate a network (Figure 2C). The KEGG functional enrichment analyses [23,24] based on significantly induced genes by LKB1 overexpression revealed that the most enriched pathways were pathways in cancer and cell cycle (Figure 2D). KEGG results also identified galactose metabolism, glycolysis/gluconeogenesis and carbohydrate digestion and absorption pathways, which are related to type II diabetes (Figure 2D). Notably, significant enrichment of several major metabolic regulatory pathways, such as the MAPK signaling pathway, Wnt signaling pathway, TGF-beta signaling pathway, p53 signaling pathway, was significantly changed by LKB1 overexpression (Figure 2D). In the MAPK signaling pathway, the mRNA levels of preadipocyte differentiation promotors (*MAP2K2*, *ATF4*, *FGF1*), preadipocyte marker (*PDGFRA*) and a cell cycle-associated protein (*CDC25B*) [25,26,27,28] were significantly inhibited by LKB1 (Figure 2E). The mRNA levels of negative adipogenic regulators (*RAC2*, *IL-1α*, *FGF2*) were induced by LKB1 (Figure 2E). In the Wnt signaling pathway, adipokines (*SFRP1*, *SFRP2*, *SFRP4*, *SFRP5*), stimulating adipocyte differentiation, were significantly reduced in LKB1-overexpressing adipocytes (Appendix A). In the TGF-beta signaling pathway, transforming growth factor-β2 (*TGFB2*), which stimulates adipocyte progenitor proliferation, and the BMP target gene *Id1*, which has critical functions in adipocyte differentiation and adipose tissue metabolism, were significantly inhibited by LKB1 overexpression (Appendix A). In the p53 signaling pathway, suppression of cytokine signaling 3 (*SOCS3*), a negative modulator of insulin signaling in adipocytes, and adipogenesis genes (*ADIPOQ*, *ACSL1*) were significantly decreased by LKB1-overexpression (Appendix A). These results indicated that LKB1 overexpression induced considerable alterations in the cell cycle and metabolic signaling pathways in SCF adipocytes.

### 3.3. LKB1 Overexpression Alters the Transcriptional Profile of Pig IMF Adipocytes

Next, we applied RNA-seq to map the transcriptional changes and adipogenic regulatory pathways in LKB1-overexpressing IMF adipocytes and controls. We identified a total of 4626 DEGs, 2201 upregulated genes and 2425 downregulated genes, which were far more than those in SCF adipocytes (Figure 3A). The mRNA level of LKB1 was dramatically elevated 2.6 times in LKB1-overexpressing IMF adipocytes compared to that in controls, which was lower than that in SCF adipocytes (Figure 3A). GO enrichment analysis of differentially expressed genes induced by LKB1 overexpression revealed pronounced changes in biological processes (cell adhesion, biological adhesion), molecular functions (microtubule-binding, transmembrane receptor protein kinase activity) and cell components (condensed chromosome, an integral component of plasma membrane) (Figure 3B and Appendix A). The cnetplot depicted the linkages of three significant biological process GO terms (cell adhesion, biological adhesion, regulation of apoptotic process) and genes involved in these terms as a network (Appendix A). KEGG functional enrichment analyses, based on significantly induced genes by LKB1 in IMF adipocytes, identified the same pathways as those in SCF adipocytes (such as MAPK signaling pathway, TGF-beta signaling pathway, p53 signaling pathway) (Figure 3C). In the MAPK signaling pathway, the mRNA levels of *PDGFRA* (down), *CDC25B* (down), *RAC2* (up), *FGF2* (up) showed the same change trends in IMF and SCF adipocytes upon LKB1 overexpression (Appendix A). However, the mRNA level of *FGF10* was significantly increased, and the mRNA levels of *MAP2K2*, *ATF4*, *FGF1* were not changed by LKB1 overexpression (Appendix A). In the TGF-beta signaling pathway, *Id1* (down) and *BMPR1A* (up) in IMF adipocytes displayed the same change trend as that in SCF adipocytes (Figure 3F). However, *GREM1* (down) had an inverse expression pattern in IMF adipocytes, and *TGFB2* and *BMP4* were not altered by LKB1 overexpression (Figure 3F). In the p53 signaling pathway, adipocyte function genes (*PCK1*, *PCK2*, *ACSL4*) were significantly increased, while adipocyte function genes (*LEPR*, *AKT2*, *JAK2*, *ISR1*) and lipogenic target genes (*PRKAA1*, *ACSL3*, *ADOPIQ*) were significantly decreased in LKB1-overexpressing IMF adipocytes (Appendix A). In addition, we also found enrichment in the PPAR signaling pathway, Toll-like receptor signaling pathway, NOD-like receptor signaling pathway, cytokine–cytokine receptor interaction, and chemokine signaling pathway in IMF adipocytes (Figure 3C). Heatmaps of cytokine–cytokine receptor interaction and chemokine signaling pathway showed that LKB1 overexpression induced considerable significant alterations in the genes encoding secreted factors, including pleiotropic cytokines (e.g., *LIF*, *CD70*, *IL10*, *IL11*), chemokines (e.g., *CCL19*, *CCL22*, *CCL2L1*, *CCL4*), growth factors (e.g., *GDF10*, *GDF11*), and transforming growth factors (e.g., *TGFB1*, *TGFB2*) (Figure 3D,E). The expression levels of *IL1R*, encoding the receptor of adipogenesis promoter IL-1Ra, and *IL11*, which inhibits adipogenesis, were significantly decreased in LKB1-overexpressed IMF adipocytes (Figure 3D). Notably, in the PPAR signaling pathway, the mRNA levels of an adipogenesis master regulator (*PPARG*), PPARG activator (*PDPK1*), lipogenesis regulators (*ADIPOQ*, *FABP4, FABP5*, *PLIN1*, *PLIN4*, *SCD*, *ACSL2*, *ME1*), fatty acid oxidation related genes (*ACADL*, *ACOX1*, *ACOX2*, *SCP2*) and a bile acid metabolism-related gene (*CYP7A*) were significantly altered by LKB1 in IMF adipocytes (Figure 3G). These results indicated that LKB1 overexpression induced similar alterations in the cell cycle and some metabolic signaling pathways in IMF and SCF adipocytes. LKB1 overexpressing IMF adipocytes also displayed enrichment in the PPAR signaling pathway, cytokine–cytokine receptor interaction and chemokine signaling pathway, which were not found in SCF adipocytes.

### 3.4. Comparison of Adipogenesis Genes of Pig IMF and SCF Adipocytes upon LKB1 Overexpression

As the above results showed that LKB1 inhibited adipogenic differentiation in both SCF and IMF adipocytes, we next compared the Log2 (fold change) values of genes involved in adipogenesis, adipogenesis regulation and fatty acid metabolism by heatmaps (Figure 4A–C). The correlation of these progression-related genes between IMF and SCF adipocytes was weak (adipogenesis: r = 0.093; adipogenesis regulators: r = −0.10; fatty acid metabolism: r = 0.16) were showed by vignette, respectively (Figure 4D–F). Transcriptional regulators of adipogenesis (*PPARG*, *CEBPB*, *CEBPD*, *ADD1*, *SREBF1*, *KLF15*) were decreased in both SCF and IMF adipocytes (Figure 4A). The TCF/LEF family (*TCF7*, *TCF7L2*) and GATA2/3 family (*GATA2*, *GATA3*), which inhibit the induction of PPARγ and C/EBPα, were increased in both SCF and IMF adipocytes (Figure 4A). Notably, adipogenesis inhibitors (*WNT10B*, *DLK1*, *GATA4*), growth factors/hormones (*TGFB1*), adipocyte secretory factors (*IL6*, *ADIPOQ*, *TNF*), and transcription factors/modulators (*RORA*, *RXRG*, *MEF2B*, *CEBPA*, *CREB1*, *NRIP1*) displayed inverse trends in IMF and SCF adipocytes upon LKB1 overexpression (Figure 4A). Some adipogenesis regulators (such as *HMG1*, *OSM*, *GDF10*, *FOXC2*, *BMP4*, *KLF5*) and fatty acid metabolism-related genes (such as *THEM5*, *ACAT1*, *ALDH2*, *ELOVL2*, *CPT1A*, *CPT1C*, *ACSL4*, *SCD5*) were also differentially regulated by LKB1 in IMF and SCF adipocytes (Figure 4B,C). In conclusion, IMF and SCF adipocytes have certain regulatory mechanisms in common, but there are also adipogenesis and fatty acid metabolism regulators in which they differ.

### 3.5. Comparison of LKB1-Induced DEGs in Pig IMF and SCF Adipocytes

To investigate the differences in the regulatory mechanism of LKB1 in IMF and SCF adipocytes, we further compared DEGs in IMF and SCF adipocytes in response to LKB1 overexpression. The Venn diagram showed the distribution of SCF- and IMF-DEGs of IMF and SCF adipocytes under LKB1 overexpression conditions (Figure 5A). KEGG results for SCF-specific DEGs (273 upregulated and 221 downregulated) revealed enrichment in pathways in cancer, Wnt signaling pathway, cell adhesion molecules (CAMs) and ubiquitin-mediated proteolysis (Figure 5B). KEGG results for IMF-specific DEGs (1641 upregulated and 1588 downregulated) revealed enrichment in cytokine–cytokine receptor interaction, chemokine signaling pathway, calcium signaling pathway and endocytosis (Figure 5C). These pathways may play distinct roles in IMF and SCF adipocytes upon LKB1 overexpression. Next, we focused on the 544 shared DEGs, and the dotplot shows the changing trends of shared DEGs (Figure 5D). Cell cycle regulators (*CCNB1*, *CDK1*, *FOXM1*), calgranulin genes (*S100A8*, *S100A12*), lipolysis regulator (*ABHD4*), E2F family (*E2F1*), Krüppel family (*KLF2*), histone chaperone proteins (*ASF1B*), transcriptional repressor (*MXD3*), CC cytokine gene (*CCL19*) and microtubule-associated protein family (*MAP2*) showed the same trend in IMF and SCF adipocytes upon LKB1 expression, suggesting that these genes and biogenic progress were similarly regulated by LKB1 in IMF and SCF adipocytes (Figure 5D). However, anti-adipogenic factors (*MMP3*, *TGM2*, *GDF10*, *TIMP3*, *WNT5A*), lipogenesis regulators (*PLIN1*, *PLIN4*, *CIDEC*), cytokines (*IL11*, *IL1A*, *IL6*, *ANGPT1*), adipocyte-released factors (*LIF*), small G protein (*RASD1*), G protein-coupled receptors (*CHRM2*, *S1PR3*), some G-protein signaling-related genes (*ADRA1B*, *FNDC1*), fibroblast growth factor family protein (*FGF10*), a positive insulin-sensitivity regulator (*NR4A2*), GTPase-activating protein (*RGS2*, *RGS4*) and mitochondrial protein (*BNIP3*) showed inverse trends in IMF and SCF adipocytes upon LKB1 expression (Figure 5D). These results suggested that the downstream targets and signaling pathways through which LKB1 functions as an adipogenesis inhibitor may be different between IMF and SCF adipocytes. Furthermore, KEGG results for shared SCF-upregulated and IMF-upregulated DEGs revealed enrichment in the regulation of actin cytoskeleton, pathways in cancer and hematopoietic cell lineage (Figure 5E). KEGG results for shared SCF-downregulated, and IMF-downregulated DEGs revealed enrichment in the cell cycle and DNA replication (Figure 5F). KEGG results for shared SCF-upregulated and IMF-downregulated or SCF-downregulated and IMF-upregulated DEGs revealed enrichment in pathways in cancer, MAPK signaling pathway and cysteine and methionine metabolism (Figure 5G). These comparative results revealed the specific and shared DEGs and related pathways in IMF and SCF adipocytes upon LKB1 expression, which may be potential targets to differentially regulate IMF and SCF adipocytes.

### 3.6. LKB1 Overexpression Inhibits Differentiation of Mouse IMF and iWAT Preadipocytes

To verify the above findings, we isolated mouse IMF and iWAT preadipocytes (Figure 6A). The flow cytometry analysis of Sca-1, a marker of adipose-derived stem cells, showed that cells separated from IMF (73.1%) contained a lower percentage of Sca-1^+^ cells than those from iWAT (98.7%) (Figure 6B). Micrographs of these separated iWAT and IMF preadipocytes after 5-day adipogenic differentiation showed numerous intracellular lipid droplets in both, but IMF mature adipocytes contained slightly more and larger lipid droplets (Figure 6C). qPCR results showed dramatic increases in the mRNA of adipogenesis (*Pparg*, *Adipoq*) and lipogenesis genes (*Scd1*, *Srebp1*) in both iWAT and IMF adipocytes during the adipogenic differentiation process (Appendix A). Consistent with the lipid droplet phenotype, IMF adipocytes expressed higher mRNA levels of *Pparg*, *Adipoq*, *Scd1* and *Srebp1* than iWAT adipocytes (Appendix A). The mRNA levels of *Lkb1*, which plays crucial roles in regulating preadipocyte growth, proliferation and differentiation via AMP-activated protein kinase (AMPK) family members [29], were significantly elevated in both iWAT and IMF adipocytes during adipogenic differentiation, and IMF adipocytes (by 1.84 times) had a lower fold increase than iWAT adipocytes (by 2.33 times) (Appendix A).

To investigate whether LKB1 has different effects on IMF and iWAT adipocytes, we generated LKB1-overexpressing IMF and iWAT adipocytes and cultured them in an adipogenic induction medium. Oil Red O staining showed that the lipid droplets in both IMF and iWAT mature adipocytes were reduced to similar degrees by LKB1 overexpression (Figure 6D). Western blot results confirmed these findings, since a clear decrease in the protein expression level of fatty acid-binding protein 4 (FABP4), a marker of lipogenesis, was observed, accompanied by a remarkable increase in the protein bands of LKB1 in both IMF and iWAT mature adipocytes (Figure 6E,F). Similarly, qPCR results showed that the mRNA level of *Lkb1* was significantly elevated in LKB1-overexpressing IMF and iWAT adipocytes, while the mRNA levels of adipogenic master regulators (*Cebpα* and *Pparg*) were significantly inhibited by LKB1 overexpression (Figure 6G,H). However, iWAT adipocytes showed larger alterations in the mRNA levels of *Lkb1*, *Cebpα* and *Pparg* (Figure 6G,H), suggesting that iWAT adipocytes were more sensitive to LKB1 overexpression. Consistently, the AMPK activator, AICAR significantly inhibited the fat droplets deposition and triacylglycerol (TG) contents in both iWAT and IMF (Appendix A). However, the sensitivity to AICAR treatment of iWAT and IMF was comparable (Appendix A).

## 4. Discussion

In this study, we compared adipogenesis-related genes in IMF and SCF adipocytes of mice and pigs upon adipogenic differentiation and LKB1 overexpression. In addition, the current study offers the first comprehensive insight into the differential role of the LKB1 gene in porcine IMF and SCF adipocytes using RNA-seq technology.

We found that IMF adipocytes exhibited slightly higher efficiency of differentiation and lipid accumulation than SCF adipocytes in both mice and pigs, with higher expression of adipogenic master regulators (*Pparg*, *Cebpa*) and lipogenesis markers (*Adipoq*, *Fabp4*) in IMF adipocytes. However, Chen et al. reported that SCF adipocytes expressed higher levels of *Pparg* and *Cebpa* during adipogenesis than IMF adipocytes [30]. In vivo, the efficiency of the differentiation of these two SVFs from intramuscular and subcutaneous adipose tissue was affected by the surrounding microenvironments, including the different concentrations of cytokines and hormones [31]. We thought that different MDI media and insulin media might influence the efficiency of differentiation of SCF and IMF adipocytes, as Chen’s study and our study differed in the MDI medium (Chen: 0.5 mM 3-isobutyl-1-methylxanthine, 0.25 μM dexamethasone, 5 mg/L insulin; our study: 0.5 mM 3-isobutyl-1-methylxanthine, 1 μM dexamethasone, 1 mg/L insulin) and insulin medium (Chen: 5 mg/L insulin; our study: 1 mg/L insulin) used for adipogenic induction. All three reagents have different and powerful effects in adipocytes, such as adipogenic differentiation, mitochondrial dysfunction and glucose metabolism [32,33,34,35]. In vivo, insulin responses of these two kinds of adipocytes, SCF and IMF adipocytes, may also differ due to the diverse physiological regulation of metabolic fluxes in iWAT and skeletal muscle [36]. It was widely accepted that ectopic adipocytes in skeletal muscle impaired insulin signaling and interacted with iWAT to induces or aggravates insulin resistance through glucose, fat metabolites or immune factors [37,38]. Our RNA-seq results revealed that different responses of adipogenesis (*WNT10B*, *DLK1*) and fatty acid metabolism (*CPT1A*, *SCD5*) regulators and insulin-related genes (*SOCS3*, *AKT2*, *IRS1*) in LKB1-overexpressing SCF and IMF adipocytes. These results suggest that SCF and IMF adipocytes may be a novel target for treating metabolic disorders, such as obesity and insulin resistance.

We found that LKB1 overexpression inhibited adipogenesis in both IMF and SCF adipocytes by inhibiting the expression levels of *Pparg* and *Cebpa*, which is consistent with previous results in 3T3-L1 adipocytes [14,15]. Insulin-sensitive *Glut4*, which mediates glucose uptake in muscle and fat, displayed a much greater decrease in IMF than SCF adipocytes upon LKB1-overexpressing. In addition, our qPCR and RNA-seq results showed that SCF adipocytes were more sensitive to LKB1 overexpression than IMF adipocytes in both mice and pigs. SCF adipocytes showed more ectopic expression of LKB1 and stronger decreases of expression levels of *Pparg* and *Cebpa* than IMF adipocytes, although per adipocyte was exposed to equal viral titers and total amounts of LKB1-overexpressing adenovirus. During adipogenic differentiation, the expression level of LKB1 in SCF adipocytes also changed to a greater extent than that in IMF adipocytes. These results suggest that the expression level of LKB1 in SCF adipocytes is more variable than that in IMF adipocytes in response to changes in the microenvironment. Notably, we also found that LKB1 expression in SVFs from mice was significantly increased at terminal differentiation, while LKB1 expression in SVFs from pigs showed a different trend after differentiation. Although all differentiated SVFs were harvested at the end of the differentiation procedures in this study and showed plenty of lipid drops and high expression levels of adipogenic regulators and lipogenesis markers, it was difficult to precisely identify the different stages of differentiation or the termination of differentiation in specific types of adipocytes. LKB1 attenuates early events of adipogenesis and responds to adipogenic cues [14]. However, the roles of LKB1 at different stages of adipocyte development remain controversial, as different studies reported various effects of LKB1 on adipogenesis transcription factors, lipolysis regulators, and browning in different adipocytes [12,15,39]. Therefore, we suspect that the various species and cell types may together influence the expression level and function of LKB1 in adipocytes. These results suggest that it is necessary to consider these influence factors on adipocytes in clinical treatment for obesity, especially visceral and gynoid obesity.

RNA-seq results revealed that IMF adipocytes showed much more specific differentially expressed genes than SCF adipocytes, suggesting that the LKB1-mediated regulatory mechanisms of adipogenesis in IMF adipocytes were more complex than those in SCF adipocytes. This phenomenon may be directly caused by the higher heterogeneity of isolated IMF adipocytes, which may contain a tiny number of muscle stem cells due to their particular location in close vicinity to muscle fibers. Furthermore, IMF adipocytes-specific DEGs were enriched in several cytokine-related pathways, suggesting that IMF adipocytes may have higher cytokine secretory activity than SCF adipocytes due to the more complex microenvironment in muscle tissue [31]. Numerous studies have indicated that cytokines secreted by both adipocytes and immune cells participate in regulating energy expenditure and adipogenesis through various pathways (such as PPARs, C/EBPs, insulin signaling and the WNT signaling pathway) or through the network formed by cytokines or unknown factors [40,41]. We assume that the abundant cytokines of skeletal muscle play critical and various roles in adipogenesis, as significant alterations in the expression levels of *IL1RA*, *IL11* and *IL34* were found in IMF adipocytes upon LKB1 overexpression. Future studies are needed to identify specific cytokines that may be targeted to differentially regulate adipogenesis in SCF and IMF adipocytes.

Comparison of transcriptome profiles of SCF and IMF adipocytes upon LKB1 overexpression revealed that the main regulatory mechanism for inhibiting adipogenesis upon LKB1 overexpression is consistent between SCF and IMF adipocytes, through inhibiting transcriptional regulators of adipogenesis (*PPARG*, *CEBPB*, *CEBPD*, *ADD1*, *SREBF1*, *KLF15*) and promoting the adipogenesis inhibitors, TCF/LEF family (*TCF7*, *TCF7L2*) and GATA2/3 (*GATA2*, *GATA3*). KEGG results also showed that regulation of the actin cytoskeleton and TGF-beta signaling pathway was upregulated in both SCF and IMF adipocytes, and cell cycle and p53 signaling pathways were downregulated. However, transcriptional regulators of adipogenesis (*CEBPA*, *KLF5*, *LEF1*, *NR1H2*, *HMGA1*, *LIF, AHR, RBL1*), regulatory signals (*BMP2*, *WNT10B*, *TGFB1*), adipogenic markers (*ADIPOQ*, *FABP4*, *PLIN1*, *PLIN2*) and fatty acid metabolism regulators (*THEM5*, *ACAT1*, *ALDH2*, *ELOVL2*, *CPT1A*, *CPT1C*, *ACSL4*, *SCD5*) had inverse change trends in SCF and IMF adipocytes upon LKB1 overexpression. Among these, the WNT signaling pathway most possibly has different roles in SCF and IMF adipocytes, as Wu et al. also found that the WNT signaling pathway was specifically enriched in SCF adipocytes treated with LGALS12-siRNA, but not IMF adipocytes [18]. KEGG results for the shared inconsistently regulated DEGs in SCF and IMF adipocytes upon LKB1 overexpression revealed enrichment in the MAPK signaling pathway, which can regulate adipogenesis at each step of the adipogenic differentiation process, from stem cells to adipocytes [42]. These results suggest that LKB1 might have different effects on immature SCF and IMF adipocytes by differentially regulating adipogenesis transcriptional regulators and signaling pathways, especially WNT signaling and MAPK signaling pathway. Metformin is an anti-diabetic drug targeting the LKB1/AMPK signaling pathway, suppressing intramuscular fat accumulation in skeletal muscle through fatty acid oxidation [43] and suppressing white adipocyte differentiation via induction of FGF21 [44]. Thus, it has directed the significance of precisely targeted treatment to understand the different effects of LKB1 on adipocytes located in different depots. In addition, several genes (such as *RGS4*, *WNT5*, *CIDEC*, *RASD1*) had inverse change trends between iWAT and TA tissues of *Adipoq-LKB1* mice, which was consistent with the findings in SCF and IMF adipocytes upon LKB1 overexpression. Triggering these potential targets may be a valuable strategy to differentially regulate adipogenesis in SCF and IMF adipocytes in vivo.

## 5. Conclusions

In conclusion, we compared LKB1-induced distinct adipogenic regulatory mechanisms between SCF and IMF adipocytes based on the DEGs and biological signaling pathways analyzed by qPCR results and transcriptomic profiles. As subcutaneous fat content is considered a negative factor associated with human health and animal economic benefits, while intramuscular fat accumulation contributes to human systemic diseases development and animal gustatory qualities, the current results would encourage further and more studies on the potential targets that could differentially control lipid deposition in subcutaneous fat and skeletal muscle, especially intramuscular cytokines and specific transcriptional regulators.

## Figures and Tables

**Figure 1 cells-09-02599-f001:**
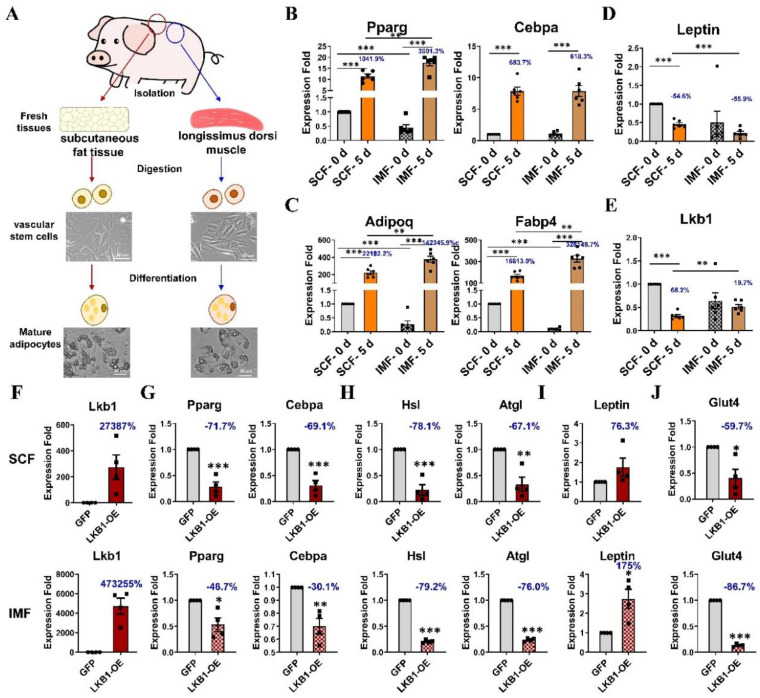
Comparison of pig intramuscular fat (IMF) and subcutaneous fat (SCF) adipocytes exposed to adipogenic differentiation medium and overexpressing liver kinase B1 (LKB1). (**A**) Experimental design. IMF and SCF preadipocytes were isolated from muscle and adipose tissues, respectively, from three 3-day-old pigs and used in subsequent experiments (n = 6). (**B**–**E**) q-PCR results of the mRNA levels of *Pparg*, *Cebpa* (**B**), *Adipoq*, *Fabp4* (**C**), *Leptin* (**D**), and *Lkb1* (**E**) in SCF and IMF adipocytes before and after adipogenic differentiation induction. (**F**–**I**) qPCR results of the mRNA levels of *Lkb1* (**F**), *Pparg*, *Cebpa* (**G**), *Hsl*, *Atgl* (**H**), *Leptin* (**I**), *Glut4* (**J**) in SCFand IMF adipocytes upon LKB1 overexpression and control adenovirus (n = 4). Data are presented as means ± SEM. * *p* < 0.05, ** *p* < 0.01, *** *p* < 0.001.

**Figure 2 cells-09-02599-f002:**
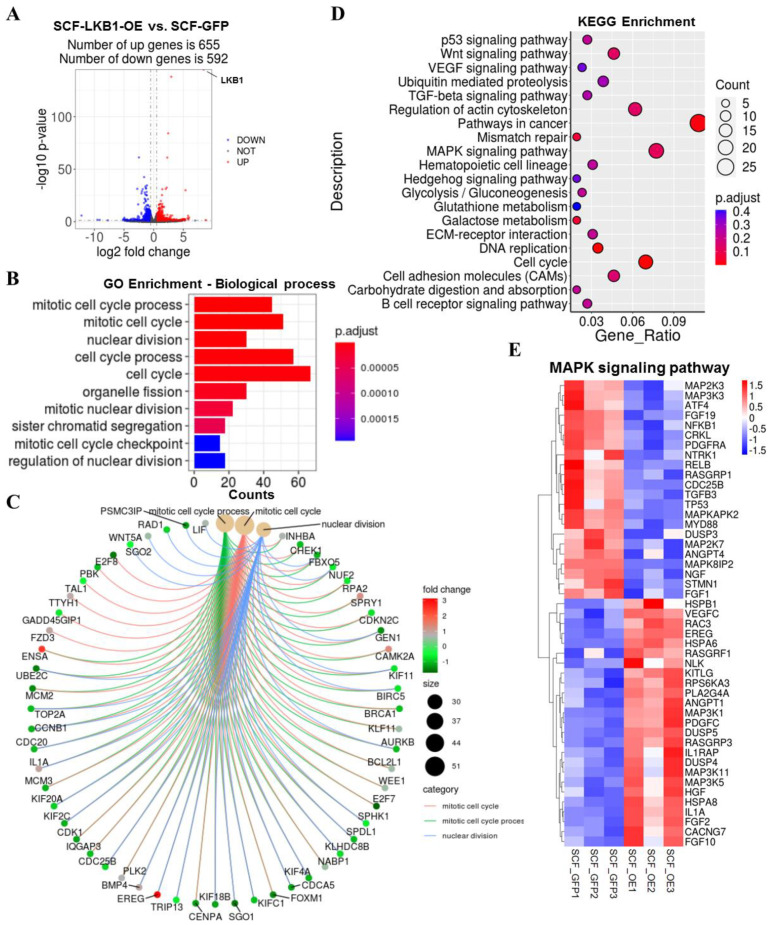
LKB1 overexpression alters the transcriptional profile of SCF adipocytes. (**A**) Log2 (fold change) values in exons of RNA-seq gene bodies in LKB1-overexpressing SCF adipocytes versus controls and the corresponding significance values displayed as -log10 (*p*-value). The transverse and vertical dotted lines indicate the cutoff value for differential expression (*p* < 0.05 and Abs (Log2 (fold changes) > 0.5). In total, 655 and 592 genes were identified that had induced (red) or repressed (blue) expression levels by cold exposure. (**B**) Gene ontology (GO) enrichment analysis based on biological processes and enriched terms were visualized by bar plot. The bar color indicates significance, and the corresponding significance values are displayed as P.adjust. The bar length indicates significantly changed gene counts of genes involved in certain categories. (**C**) The cnetplot depicts the linkages of the three most enriched GO terms (mitotic cell cycle process, mitotic cell cycle, nuclear division) and genes involved in these terms as a network. The yellow dots indicate enriched GO terms, and the size of each dot indicates gene counts involved in certain GO terms. The smaller dots indicate genes involved in these terms. The color of each smaller dot indicates genes with log2(fold change) values in LKB1-overexpressing SCF adipocytes versus controls. (**D**) Functional enrichment analyses using Kyoto Encyclopedia of Genes and Genomes (KEGG) pathways. The triangle size indicates gene counts. The dot color indicates significance, and the corresponding significance values are displayed as P.adjust. (**E**) Heatmap of TPM expression values genes involved in the MAPK signaling pathway. Only genes with *p* < 0.05 are displayed.

**Figure 3 cells-09-02599-f003:**
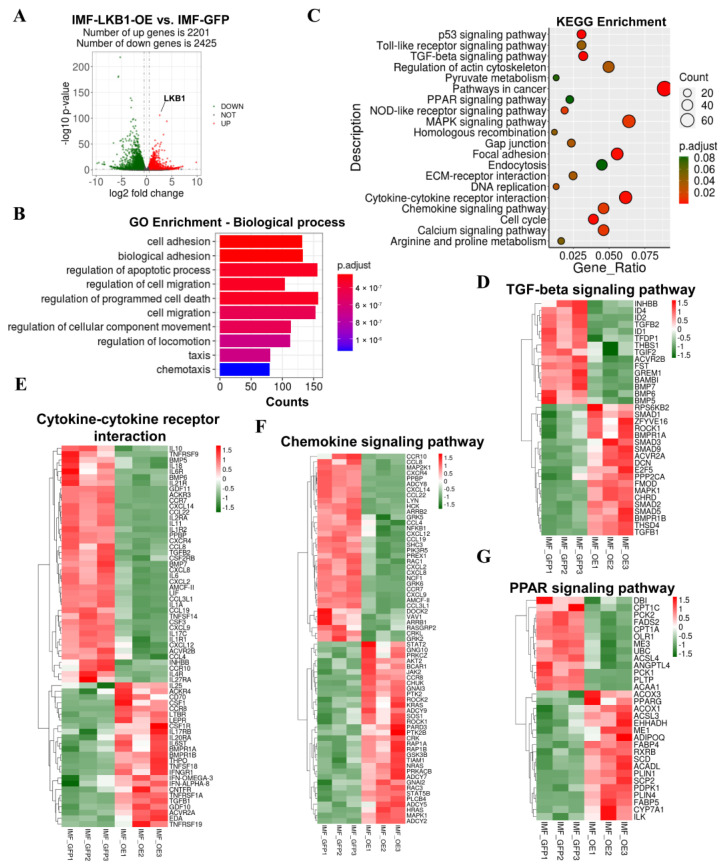
LKB1 overexpression alters the transcriptional profile of IMF adipocytes. (**A**) Log2 (fold change) values in exons of RNA-seq gene bodies in LKB1-overexpressing IMF adipocytes versus controls and the corresponding significance values displayed as -log10 (*p*-value). The transverse and vertical dotted lines indicate the cutoff value for differential expression (*p* < 0.05 and Abs (log2 (fold changes)) > 0.5). In total, 2201 and 2425 genes were identified that had induced (red) or repressed (green) expression levels by cold exposure. (**B**) Gene ontology (GO) enrichment analysis based on biological processes and enriched terms were visualized by bar plot. The bar color indicates significance, and the corresponding significance values are displayed as P.adjust. The bar length indicates significantly changed gene counts of genes involved in certain categories. (**C**) Functional enrichment analyses using Kyoto Encyclopedia of Genes and Genomes (KEGG) pathways. The triangle size indicates gene counts. The dot color indicates significance, and the corresponding significance values are displayed as P.adjust. (**D**–**G**) Heatmap of TPM expression values of genes involved in cytokine–cytokine receptor interaction (**D**), chemokine signaling pathway (**E**), TGF-beta signaling pathway (**F**), and PPAR signaling pathway (**G**). Only genes with *p* < 0.05 are displayed.

**Figure 4 cells-09-02599-f004:**
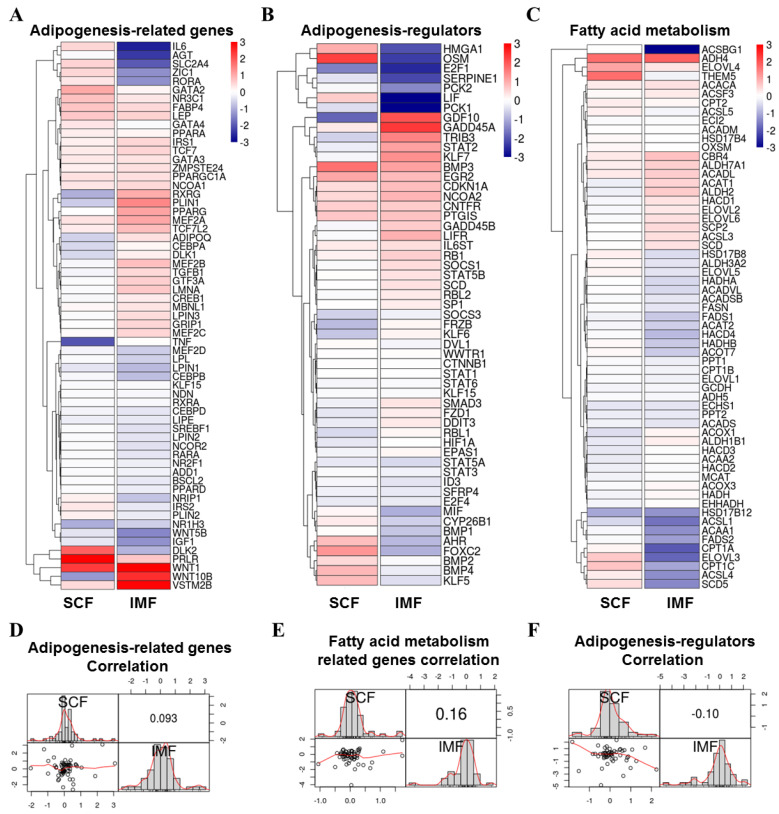
Comparison of adipogenesis genes of IMF and SCF adipocytes upon LKB1 overexpression. (**A**–**C**) Heatmap of log2 (fold change) values of adipogenesis-related genes (**A**), adipogenic-regulators (**B**) and fatty acid metabolism-related genes (**C**) in IMF and SCF adipocytes upon LKB1 overexpression. (**D**–**F**) Correlation matrix of IMF and SCF adipocytes based on Pearson’s correlation coefficient for the subset of genes involved in adipogenesis-related genes (**D**), adipogenesis regulators (**E**) and fatty acid metabolism (**F**), respectively.

**Figure 5 cells-09-02599-f005:**
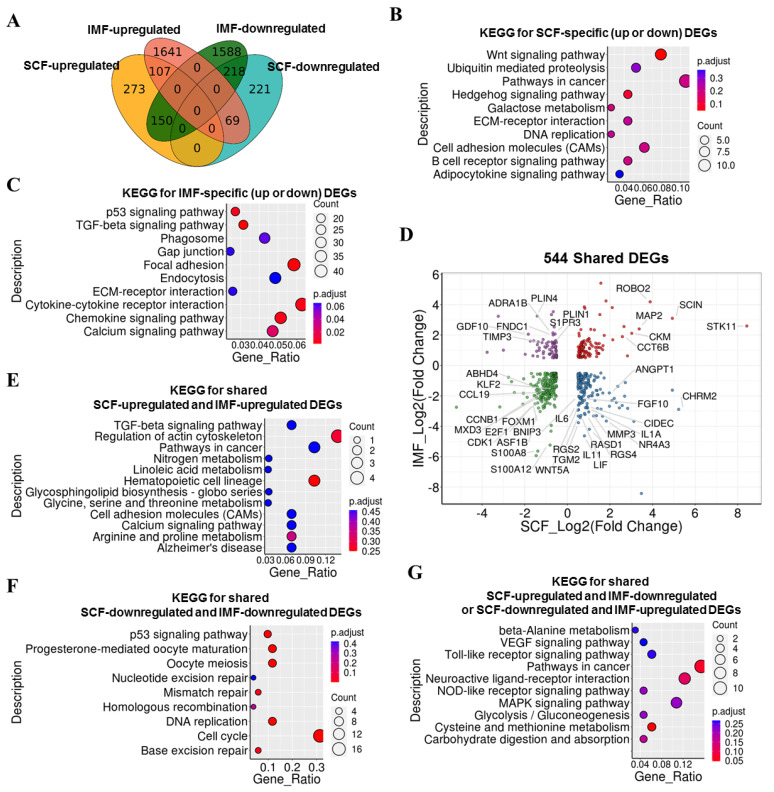
Comparison of LKB1-induced DEGs in IMF and SCF adipocytes. (**A**) Venn diagram of LKB1-induced DEGs in IMF and SCF adipocytes, which were further divided into upregulated DEGs in SCF adipocytes, downregulated DEGs in SCF adipocytes, upregulated DEGs in IMF adipocytes, downregulated DEGs in IMF adipocytes. (**B**,**C**) Functional enrichment analyses using KEGG pathways of SCF adipocyte-specific DEGs (273 upregulated DEGs and 221 downregulated DEGs) (B) and IMF adipocyte-specific DEGs (1641 upregulated DEGs and 1588 downregulated DEGs) (**C**). (**D**) Dotplot of 544 shared DEGs in IMF and SCF adipocytes upon LKB1 overexpression. The *x*-axis indicates log2 (fold change) values of genes in SCF adipocytes, and the *y*-axis indicates log2 (fold change) values of genes in IMF adipocytes. The shared DEGs were divided into 4 categories: upregulated in both IMF and SCF adipocytes (red), downregulated in both IMF and SCF adipocytes (green), upregulated in IMF adipocytes and downregulated in SCF adipocytes (blue), and downregulated in IMF adipocytes and upregulated in SCF adipocytes (purple). (**E**–**G**) Functional enrichment analyses using KEGG pathways of upregulated shared DEGs in both IMF and SCF adipocytes (**E**), downregulated shared DEGs in both IMF and SCF adipocytes (**F**), and inversely regulated shared DEGs (**G**).

**Figure 6 cells-09-02599-f006:**
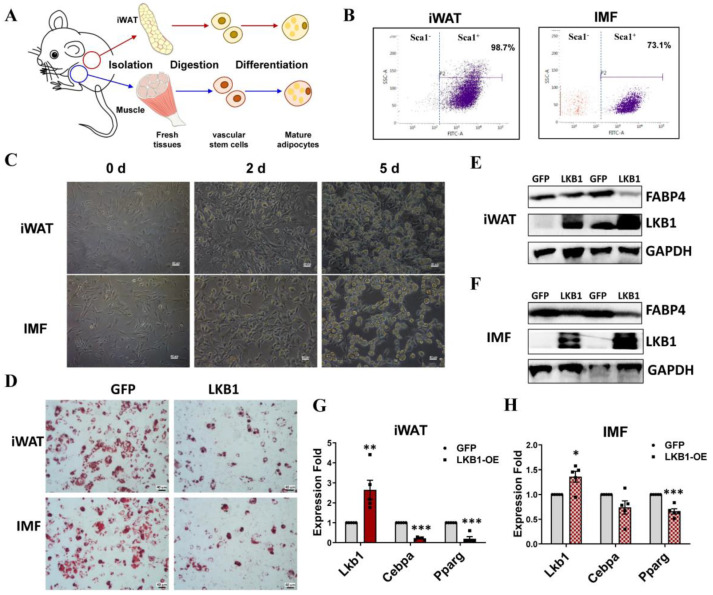
Comparison of mouse IMF and inguinal white adipose tissue (iWAT) adipocytes during adipogenic differentiation and upon LKB1 regulation. (**A**) Experimental design. IMF and iWAT preadipocytes were isolated from muscle and adipose tissue, respectively, from growing mice and used in subsequent experiments. (**B**) Flow cytometry-based quantitation of Sca1^+^ populations in iWAT and IMF adipocytes. (**C**) Optical microscope images of iWAT and IMF adipocytes during adipogenic differentiation. (**D**) Oil Red O-staining of iWAT and IMF adipocytes upon LKB1 overexpressing and control adenovirus infections. (**E**,**F**) Western blot results of the protein expression levels of FABP4, LKB1 and GAPDH in iWAT (**E**) and IMF (**F**) adipocytes upon LKB1-overexpressing and control adenovirus infections. (**G**,**H**) qPCR results of the mRNA levels of *Lkb1*, *Cebpa*, and *Pparg* in iWAT (**G**) and IMF (**H**) adipocytes upon LKB1-overexpressing and control adenovirus infections. Data are presented as means ± SEM (n = 5). * *p* < 0.05, ** *p* < 0.01, *** *p* < 0.001.

## Data Availability

All original/source data in the current study are available from the corresponding author on request.

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
