# Peer review of "LKB1 Differently Regulates Adipogenesis in Intramuscular and Subcutaneous Adipocytes through Metabolic and Cytokine-Related Signaling Pathways"

_cells, 2020, doi:10.3390/cells9122599_

Round 1
Reviewer 1 Report
In their manuscript entitled: "LKB1 differently regulates adipogenesis in intramuscular and subcutaneous adipocytes through metabolic and cytokine-related signaling pathways", Zyye Xu et al. explored the expression level of genes implicated in the regulatory mechanism of LKB1 in cultured adipocytes dissected from the subcutaneous and intramuscular fat of two different animal models, pig and mouse. The subject is interesting and the experimental design is clearly described. However, since the complexity of the LKB1 downstream pathways, the authors need to speculate more the consequence of LKB1 overexpression in the context of the biology functions. Introduction and conclusion paragraphs need to be improved. Also, in the result sections the authors have to deep explain what each data suggest.
The authors need to address the following questions.
1. In the Materials and Methods section, (paragraph 2.1), it is not clear to me how many animals were used in the study. In the line 70 (...were harvested from three 3-day-old male) the authors mean they used three pigs of 3-days old. Later, there is some incongruence reported in the legend of figure 1 that needs to be clarified. In the Line 197 the authors say n=6 experiments were performed. What does n=6 represent?
2. How many days the cell were maintained in culture before to be processed? Add this information to the methods.
3. Add references to the paragraphs 2.3, 2.4, 2.6 and 2.8
4. Also, while in the graphics 1B, 1C, 1D and 1E the data are reported as Expression Fold ( which is an arbitrary unit) in the fig 1F,G, H they are reported the Expression Folder as %. Graphics need to be consistent.
5. In the figure 6, scale bar are missed in C and D. The text is missed in the left Y axis of G and H.
Author Response
Point 1: In the Materials and Methods section, (paragraph 2.1), it is not clear to me how many animals were used in the study. In the line 70 (...were harvested from three 3-day-old male) the authors mean they used three pigs of 3-days old. Later, there is some incongruence reported in the legend of figure 1 that needs to be clarified. In the Line 197 the authors say n=6 experiments were performed. What does n=6 represent?
Response 1: We have corrected the incongruence in the legend of figure 1 (Line 199) and n=6 represents the repeat number of harvested cell wells which were applied for RNA extraction and qPCR.
Point 2: How many days the cell were maintained in culture before to be processed? Add this information to the methods.
Response 2: We have added this information to the methods (Line 84).
Point 3: Add references to the paragraphs 2.3, 2.4, 2.6 and 2.8
Response 3: We have added references to the paragraphs 2.3, 2.4, 2.6 and 2.8 (Line 99, 107, 131, 157).
Point 4: Also, while in the graphics 1B, 1C, 1D and 1E the data are reported as Expression Fold ( which is an arbitrary unit) in the fig 1F,G, H they are reported the Expression Folder as %. Graphics need to be consistent.
Response 4: We have modified the graphics in figure 1 (Figure 1B, C, D, E, G, H) and the descriptions in the result (Line 174, 175, 178, 182, 187, 192, 194).
Point 5: In the figure 6, scale bar are missed in C and D. The text is missed in the left Y axis of G and H.
Response 5: We have added missed scale bar are missed in C and D of figure 6 (Figure 6C, D). We have added the missed text in the left Y axis of G and H (Figure 6G, H).
Reviewer 2 Report
Xu et al. collected stromal vascular cells from subcutaneous fat (SCF) and intramuscular fat (IMF) of pig, and infected with GFP-, or LKB1-overexpressing adenovirus, followed by differentiation in to adipocytes in vitro. They performed transcriptome analysis in these cells.
LKB1 overexpression inhibited adipogenesis in both SCF and IMF adipocytes.
Inhibition rate of adipocyte marker genes by LKB1 overexpression were larger in SCF-derived adipocytes than IMF-derived ones.
Transcriptomics results showed that IMF adipocytes had many more differentially expressed genes than SCF adipocytes.
Notably, adipogenesis inhibitors (WNT10B, DLK1, GATA4), growth factors/hormones (TGFB1), adipocyte secretory factors (IL6, ADIPOQ, TNF), and transcription factors/modulators (RORA, RXRG, MEF2B, CEBPA, CREB1, NRIP1) displayed inverse trends in IMF and SCF adipocytes upon LKB1 overexpression.
Some of adipogenesis regulators (such as HMG1, OSM, GDF10, FOXC2, BMP4, KLF5) and fatty acid metabolism related genes (such as THEM5, ACAT1, ALDH2, ELOVL2, CPT1A, CPT1C, ACSL4, SCD5) were also differentially regulated by LKB1 in IMF and SCF adipocytes.
Following concerns should be properly addressed to make the paper robust.
1)Authors clearly indicated that overexpression of LKB1 had different effects in adipocytes derived from SCF or IMF.
It is very important to compare protein levels of LKB1 between SCF and IMF. Much protein levels of LKB1 might be the reason for reduced adipocyte differentiation in SCF.
Author should compare LKB1 protein levels among SCF and IMF infected with GFP or LKB1 viruses in Fig1.
2)It is very interesting that Adipoq expression was reduced in SCF. but elevated in IMF by LKB1 overexpression.
Author should perform ELISA or Western blotting of Adipoq among groups.
Adipoq expression is very high, and easy to detect.
3)It is important to reveal the mechanism of different response of LKB1 in SCF and IMF.
LKB1 activates AMPK, therefore it should be informative to show the effect of AMPK activator in SCF or IMF.
AMPK activator, such as AICAR, might have similar effects as LKB1 overexpression virus.
Authors should show the effect of AMPK activator on adipocyte differentiation in SCF and IMF.
4)Authors should show the microscopic image of differentiated adipocytes in Fig1.
Reduction of adipocyte marker gene expression may indicate reduced adipocyte number or reduced gene expression per adipocytes.
Author Response
Point 1: Authors clearly indicated that overexpression of LKB1 had different effects in adipocytes derived from SCF or IMF. It is very important to compare protein levels of LKB1 between SCF and IMF. Much protein levels of LKB1 might be the reason for reduced adipocyte differentiation in SCF. Author should compare LKB1 protein levels among SCF and IMF infected with GFP or LKB1 viruses in Fig1.
Response 1: We added the LKB1 mRNA levels in SCF and IMF infected with GFP or LKB1 viruses in Fig 1F (Figure 1F, Line 186~188).
Point 2: It is very interesting that Adipoq expression was reduced in SCF. but elevated in IMF by LKB1 overexpression. Author should perform ELISA or Western blotting of Adipoq among groups. Adipoq expression is very high, and easy to detect.
Response 2: Sorry, we do not have enough SCF and IMF primary cells to carry out this experiment.
Point 3: It is important to reveal the mechanism of different response of LKB1 in SCF and IMF. LKB1 activates AMPK, therefore it should be informative to show the effect of AMPK activator in SCF or IMF. AMPK activator, such as AICAR, might have similar effects as LKB1 overexpression virus. Authors should show the effect of AMPK activator on adipocyte differentiation in SCF and IMF.
Response 3: We have showed the effect of AMPK activator, AICAR, on adipocyte differentiation in SCF and IMF of mice in Supplementary figure 3D (Supplementary Figure 3D, E Line 418~421).
Point 4: Authors should show the microscopic image of differentiated adipocytes in Fig1.
Reduction of adipocyte marker gene expression may indicate reduced adipocyte number or reduced gene expression per adipocytes.
Response 4: We have showed the microscopic image of differentiated adipocytes in Fig1 (Figure 1A).
Round 2
Reviewer 2 Report
Authors responded to all of raising issues, and now I have no further concerns.